# Learning Curve in Robotic-Assisted Total Knee Arthroplasty: A Systematic Review of the Literature

Giorgio Cacciola [1], Francesco Bosco [1], Fortunato Giustra [1], Salvatore Risitano [1], Marcello Capella [1], Alessandro Bistolfi [2], Alessandro Massè [1] and Luigi Sabatini [1,*]

[1] Department of Orthopedics and Traumatology, University of Turin, CTO Turin, Via Zuretti 29, 10126 Turin, Italy
[2] Orthopaedics and Traumatology, Ospedale Cardinal Massaia Asti, Via Conte Verde 125, 14100 Asti, Italy
[*] Correspondence: luigisabatini.ort@gmail.com

**Abstract:** Several innovations have been introduced in recent years to improve total knee arthroplasty (TKA). Robotic-assisted surgery is gaining popularity for more precise implant placement while minimizing soft tissue injury. The main concerns are increased cost, operative time, and a significant learning curve. This systematic review aims to analyze the surgical time learning curve, implant placement accuracy, and complications related to robotic-assisted TKA (raTKA). A systematic literature review was performed according to the Preferred Reporting Items for Systematic Reviews and Meta-Analyses guidelines. The research was conducted up to September 2022 in four databases (PubMed/MEDLINE, Embase, Scopus, and the Cochrane Database of Systematic Reviews), with the following key terms: "robotic-assisted", "total knee arthroplasty", "robotic", "TKA", "learning", and "TKR". The methodology quality of the studies was assessed using the Methodological Index for Non-Randomized Studies (MINORS) criteria. This systematic review was registered on the International Prospective Register of Systematic Reviews (PROSPERO), ID: CRD42022354797, in August 2022. Fifteen clinical studies that analyzed the raTKA learning curve of 29 surgeons and 2300 raTKAs were included in the systematic review. Fourteen surgeons reported the presence of an inflection point during the learning curve. Few studies have reported the learning curve of raTKA regarding lower limb alignment, component position, and intraoperative and postoperative complications. The main finding of this systematic review is that the procedure number required to reach the learning plateau is about 14.9 cases. Furthermore, an average decrease in surgical time of 23.9 min was described between the initial and proficiency phases; the average surgical time in the two phases was 98.8 min and 74.4 min, respectively. No learning curve was observed for implant placement and lower limb alignment because the implants were correctly placed from the first raTKAs. No significant complication rates were reported during the raTKA learning curve.

**Keywords:** TKA; total knee arthroplasty; knee replacement; orthopedic; robotic; robotic-assisted; raTKA; learning; curve; limb alignment

## 1. Introduction

Total knee arthroplasty (TKA) is a safe and successful surgical procedure for end-stage knee osteoarthritis treatment [1,2]. More than half a million surgeries are performed annually in the United States, and epidemiological studies predict a more than 650% increase by 2030 [3]. Despite high long-term survival of more than 90% after ten years, about 20% of patients are dissatisfied with their surgeries [1–4].

In recent years, several innovations have been introduced into clinical practice to improve TKA outcomes, such as upgraded materials, minimally invasive surgery, innovative TKA alignment, computer-assisted surgery (CAS), intraoperative navigation, augmented reality, and robotic-assisted (ra) surgery [5–11]. Ra surgery has fascinated many since its beginnings around 1980. Ra procedures allow surgeons to perform complex operations

with greater precision, flexibility, and control than conventional techniques and have become a staple in operating rooms worldwide. In recent decades, considerable technological advances have made robotics applications possible in multiple surgery fields, including orthopedics [12]. Robotic-assisted TKA (raTKA) has gained popularity, and its use is steadily growing [13,14]. Despite its theoretical advantages, raTKA raises some concerns, such as increased cost and operative time, absence of long-term follow-up studies, and a significant learning curve to refine the technique [8,14]. A not-yet well-defined learning curve is one of the major concerns of surgeons approaching raTKA [15]. The learning curve has a key role in training surgeons by allowing improved performance, accuracy, and reduced surgical time [16]. It is characterized by three phases: a rapid improvement through the first cases, a successive modest but steady improvement achieved with increased experience, and finally, a plateau phase where additional experience does not influence improvement. The "inflection point" is defined as the number (n) of procedures required to achieve proficiency in each stage of the surgical procedure [16,17]. Identifying the learning curve of raTKA has several clinical applications for patient safety, surgeon training, and cost-effectiveness.

The primary purpose of this systematic review is to analyze the surgical time learning curve of raTKA. The secondary aims are to investigate the learning curve associated with implant placement accuracy and complications related to raTKAs.

## 2. Materials and Methods

A systematic literature review was performed according to the Preferred Reporting Items for Systematic Reviews and Meta-Analyses (PRISMA) guidelines [18]. The study was registered in the International Prospective Register of Systematic Reviews (PROSPERO), ID: CRD42022354797, in August 2022 [19,20].

### 2.1. Study Design and Search Strategy

Literature research in four databases (PubMed/MEDLINE, Embase, Scopus, and the Cochrane Database of Systematic Reviews) was performed until September 2022. The research was focused on studies dealing with the learning curve of raTKA. The following key terms were used in association with the Boolean operators "AND" and "OR": "robotic-assisted", "total-knee-arthroplasty", "robotic", "TKA", "learning", and "TKR".

### 2.2. Study Screening

The initial database search resulted in 647 studies. Two independent reviewers screened the titles and abstracts of the studies identified through the comprehensive literature analysis (GC and FB). Duplicate articles were removed. The full text was reviewed in case of title/abstract analysis discrepancies. In case of discrepancies, a third author (FG) was consulted. After applying the inclusion and exclusion criteria, 15 clinical studies [21–35] investigating the learning curve in robotic-assisted total knee arthroplasty were included in this systematic review. Furthermore, a cross-reference check for the inclusion of possibly relevant studies was performed. The PRISMA flowchart is shown in Figure 1.

### 2.3. Inclusion and Exclusion Criteria

Clinical studies on raTKA that respected the following criteria were included: studies on human subjects with levels of evidence (LoE) from 1 to 4 as defined by the Oxford Centre for Evidence-Based Medicine 2011 and clinical studies written in English. No publication time restrictions were established. Preclinical studies, book chapters, editorials, reviews, technical reports, and abstracts of scientific meetings were excluded.

### 2.4. Data Extraction and Collection

Two different reviewers (GC and FB) independently collected data from included studies in a template with the following characteristics: demographics data, data on the operative time learning curve, implant accuracy, and complications.

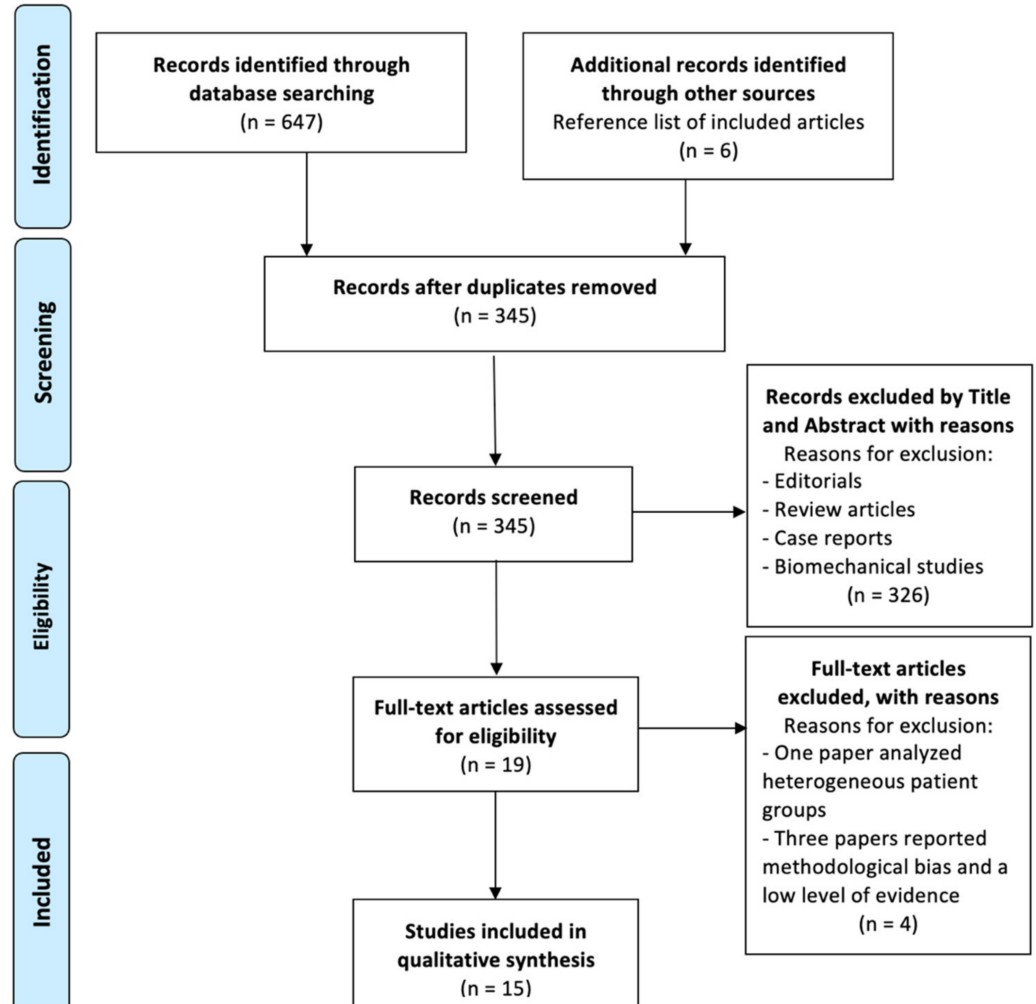

**Figure 1.** Preferred Reporting Items for Systematic Reviews and Meta-Analyses (PRISMA). Flowchart of studies included in this systematic review.

*2.5. Quality Assessment*

The articles were analyzed using the Levels of Evidence (LoE) defined by the Oxford Centre for Evidence-Based Medicine 2011. Articles were graded from I to V, where LoE I indicates the best study design [36]. The methodological quality of the included studies was evaluated by two authors using the Methodological Index for Non-Randomized Studies (MINORS) criteria. A third author resolved cases of disagreement [37,38].

**3. Results**

Based on the analysis of the included studies, it was found that there were two LoE II studies [26–33], six LoE III studies [21,24,25,27,34,35], and seven LoE IV studies [22–28,32]. The mean MINORS score for non-comparative studies was 9.3 (range 6–12) and 15.5 for comparative studies (range 14–17) (Table 1).

**Table 1.** Main demographic characteristics and surgical data.

| Author and Publication Year | LoE | Study Design | raTKA | mTKA | Surgeons N° | Expertise | Age Mean (y.o.) | M/F | BMI Mean (kg/m²) | Minors Score |
|---|---|---|---|---|---|---|---|---|---|---|
| Schopper et al. (2022) [21] | III | Prospective | Mako (Stryker) | Triathlon (Stryker) | 3 | 1 experienced surgeon, 2 non-experienced | 71 | 37/18 | 30.8 | 12 |
| Patel et al. (2022) [22] | IV | Retrospective | Mako (Stryker) | Triathlon (Stryker) | 1 | 1 fellowship-trained surgeon | 64.2 | 353/251 | 33 | 6 |
| Vanlommel et al. (2021) [23] | IV | Retrospective | ROSA (Zimmer) | Persona PS (Zimmer) | 3 | 3 high volume surgeons | 68.7 | 46/44 | 31.3 | 9 |
| Thiengwittayaporn et al. (2021) [24] | III | RCT | Navio (Smith & Nephew) | Legion (Smith & nephew) | 1 | 1 senior surgeon | 69 | giu-69 | 28 | 13 |
| Savov et al. (2021) [25] | III | Prospective | Navio (Smith & Nephew) | Journey II (Smith & Nephew) | 1 | 1 senior surgeon | 64.4 | 22/48 | 28.8 | 14 |
| Mahure et al. (2021) [26] | II | Prospective | Think Surgical | Persona PS (Zimmer), Corin CR, and PS (Unity Knee) | 4 | 4 fellowship-trained recon surgeons | 65.9 | 58/57 | 30.7 | 11 |
| Bell et al. (2021) [27] | III | Prospective | Navio (Smith & Nephew) | Journey II (Smith & Nephew) | 1 | 1 fellowship-trained surgeon | N/A | N/A | N/A | 17 |
| Ali et al. (2021) [28] | IV | Retrospective | Mako (Stryker) | N/A | 2 | 2 non-fellowship surgeons | 66.1 | 50/70 | 33.5 | 13 |
| Vermue et al. (2020) [29] | IV | Retrospective | Mako (Stryker) | Triathlon (Stryker) | 6 | 2 low volume, 1 medium volume, and 3 high volume surgeons | 70.4 | 252/134 | 30 | 18 |
| King et al. (2020) [30] | IV | Retrospective | Mako (Stryker) | Triathlon (Stryker) | 1 | 1 senior surgeon | 68 | 71/131 | N/A | 14 |
| Collins et al. (2020) [31] | IV | Retrospective | Navio (Smith & Nephew) | Legion (Smith & nephew) | 1 | 1 senior surgeon | 67 | 36/36 | N/A | 6 |
| Naziri et al. (2019) [32] | IV | Retrospective | Mako (Stryker) | Triathlon (Stryker) | 1 | 1 fellowship-trained surgeon | 69.5 | 16/24 | 29.1 | 9 |
| Kayani et al. (2018) [33] | II | Prospective | Mako (Stryker) | Triathlon (Stryker) | 1 | N/A | 68.7 | 27/33 | 26.1 | 17 |
| Garau et al. (2019) [34] | III | Prospective | Mako (Stryker) | N/A | 1 | 1 senior surgeon | N/A | N/A | N/A | 12 |
| Sodhi et al. (2018) [35] | III | Retrospective | Mako (Stryker) | N/A | 2 | 2 lower limb certified surgeons | N/A | N/A | N/A | 6 |

LoE: Oxford Centre for Evidence-Based Medicine 2011 Levels of Evidence; raTKA: robotic-assisted TKA; mTKA: manual TKA: N°: number of evaluation cases; y.o.: years old; M: male; F: female; BMI: Body Mass Index; kg: kilograms; m²: square meter; PS: Posterior-stabilized; CR: cruciate-retaining; N/A: not available; RCT: Randomized control trial.

### 3.1. Demographic Characteristics and Surgical Data

This study analyzed the raTKA learning curve of 29 surgeons (surgeons' competencies are listed in Table 1). A total of 2300 raTKAs were included in the final analysis. The mean age of the treated patients was 67.1 (64.2–71) years, 50.7% were women, and the mean BMI was 30.3 (26.1–33.5) kg/m$^2$. Nine studies used MAKO (Stryker, n: 839 raTKA) [21,22,28–30,32–35], four studies used NAVIO (Smith & Nephew, n: 264 raTKA) [24,25,27,31], one study used Think Surgical (Think Surgical Robot, n:108 raTKA) [26], and one study used ROSA (Zimmer, n: 90 raTKA) [23]. Triathlon (Stryker, n: 1347 RATKA) was the most widely used TKA [21,22,29,30,32,33], followed by Legion (Smith & Nephew, n: 147 RATKA) [24,31], Journey II (Smith & Nephew, n: 117 RATKA) [25,27], Persona (Zimmer, n: 90 RATKA) [26], and Corin (Unity Knee, n: 107 RATKA) [26]. Implants were not specified in three studies (n: 492 RATKA) [28,34,35]. Fourteen studies reported the surgeon's experience [21–32,34,35]. There were nine fellowship-trained surgeons (31%), six "high-volume" (>200 TKA per year) surgeons (20.7%), four surgeons with no previous computer- or robot-assisted TKA experience (13.8%), two surgeons with previous computer- or robot-assisted TKA experience (6.9%), one medium-volume (100–200 TKA per year) surgeon (3.5%), and one low-volume (<100 TKA per year) surgeon (3.5%). In five cases, the surgeon's experience was not reported (17.3%). The main demographic and surgical characteristics are reported in Table 1.

### 3.2. Surgical Time

Fourteen surgeons reported the presence of an inflection point during the learning curve. The mean inflection point was observed after 14.9 cases (5–43 cases). Two studies [22,34] reported that the inflection point was reached within the first 200 cases [22] and within the first 40 cases [34]. Five studies reported average surgical times between the learning and proficiency phases [23–25,29,33]. Three studies reported that the average surgical time was 98.3 min for the learning phase and 74.4 min for the proficiency phase. Seven studies compared the average surgical time between the first and last cases of raTKA [22–24,27,28,34,35]. In all but one case, the average surgical time for the first cases was significantly longer than that of the subsequent cases. The only exception was reported by one of the two surgeons in the study by Ali et al. [28]. They reported no improvement in the average surgical time between the first and last cases of raTKA. Five studies reported a statistically superior average surgical time for the first cases of raTKA patients compared to the average surgical time of manual TKA (mTKA) [24,29,32,34,35]. The surgical time results are reported in Table 2.

**Table 2.** Learning curve and surgical time of raTKA.

| Author and Publication Year | raTKA N° | Inflection Point | Learning Phase Mean Surgical Time (min) | Proficiency Phase Mean Surgical Time (min) | *p*-Value | Improvement in Surgical Time between the First and Last Cases of raTKA | Improvement in Surgical Time of raTKA Compared to mTKA |
|---|---|---|---|---|---|---|---|
| Schopper et al. (2022) [21] | 55 | No inflection for two surgeons; after 5 cases for the third surgeon | N/A | N/A | N/A | N/A | N/A |
| Patel et al. (2022) [22] | 604 | Reached within the first 200 cases | N/A | N/A | N/A | Significant improvement (*p* < 0.05) between the first and last cases (84.9 min vs. 62 min) | N/A |
| Vanlommel et al. (2021) [23] | 90 | After 10, 6, and 11 cases, respectively, for the three surgeons | 102.4 | 86.5 | <0.05 | The average surgical time of the first cases was significantly higher than the last cases | N/A |
| Thiengwittayaporn et al. (2021) [24] | 75 | After 7 cases | 100.7 | 67.4 | <0.05 | Significant decrease in mean surgical time between the first and last cases (49.1 min vs. 36.5 min) | The mean surgical time of all raTKA was significantly longer than that of mTKA |
| Savov et al. (2021) [25] | 59 | After 11 cases | No difference in average surgical time before and after the inflection point | | N/A | N/A | N/A |
| Mahure et al. (2021) [26] | 107 | No inflection points for one surgeon. After 12, 16, and 19, respectively, for the remaining 3 surgeons | N/A | N/A | N/A | N/A | N/A |
| Bell et al. (2021) [27] | 58 | After 29 cases | N/A | N/A | N/A | Significant decrease in mean surgical time between the first and last cases (41.8 min vs. 31.1 min). | N/A |
| Ali et al. (2021) [28] | 120 | N/A | N/A | N/A | N/A | A significant decrease between the first and last cases was reported for one surgeon, but no difference for the second surgeon | N/A |
| Vermue et al. (2020) [29] | 386 | After 11, 22, and 43 cases for high-volume surgeons. No inflection points for medium and low-volume surgeons. | Statistically significant lower time for high-volume surgeons during the proficiency phase compared to the learning phase | | N/A | N/A | The mean surgical time in raTKA compared to mTKA was longer for the first 10 cases. No differences were reported between the last cases of raTKA and mTKA |
| Kayani et al. (2018) [33] | 60 | After 7 cases | 89.2 | N/A | <0.05 | N/A | N/A |

**Table 2.** *Cont.*

| Author and Publication Year | raTKA N° | Inflection Point | Learning Phase Mean Surgical Time (min) | Proficiency Phase Mean Surgical Time (min) | *p*-Value | Improvement in Surgical Time between the First and Last Cases of raTKA | Improvement in Surgical Time of raTKA Compared to mTKA |
|---|---|---|---|---|---|---|---|
| Garau et al. (2019) [34] | 132 | Reached within the first 40 cases | N/A | N/A | N/A | The average surgical time in the first cases was higher than in the last cases | For one surgeon, all patients treated with raTKA showed higher mean surgical times than mTKA. For the second surgeon, the mean surgical time was significantly higher for the first 20 cases only |
| Sodhi et al. (2018) [35] | 240 | N/A | N/A | N/A | N/A | The average surgical time for both surgeons was higher in the first group of patients than in the later groups | The average surgical time for the first cases was higher than for mTKA, but no difference was noted for the last cases |
| Naziri et al. (2019) [32] | 40 | N/A | N/A | N/A | N/A | N/A | The mean surgical time for all raTKA was significantly higher (*p* < 0.05) than for mTKA. No difference in mean surgical time was reported from the last 20 cases of raTKA and mTKA |

raTKA: robotic-assisted TKA; N°: number of evaluation cases; min: minutes; mTKA: manual TKA; N/A: not available; *p*: *p*-value; vs.: versus.

*3.3. Lower Limb Alignment*

Four articles found that the learning curve was not associated with implant position [25,26,29,33]. Schopper et al. found a high percentage of outliers in mTKA compared to raTKA after reaching the inflection point [21]. The difference between intraoperative and postoperative measurements was reported in three studies [21,25,29]. One study reported no difference between the two measurements [21]. One study reported that medial proximal tibial angle (MPTA), lateral distal femoral angle (LDFA), and hip–knee–ankle angle (HKA) differed significantly between intraoperative and postoperative measurements [25]. Finally, one study reported that the mean postoperative HKA was 1.2° more valgus than that measured intraoperatively [29]. Three studies reported differences in lower limb alignment between raTKA and mTKA [24,32,33]. Two articles reported significantly better lower limb alignment in the raTKA cohort [24,33]. In contrast, one study reported no difference in lower limb alignment between raTKA and mTKA [32]. The incidence of outliers (mispositioning greater than 3° from planned positioning) was reported by six studies [21,24–26,31,32]. Two studies [21,32] did not report any outliers in their cohorts of patients undergoing raTKA. One study [24] reported that the outlier rate was lower in raTKA than in mTKA. One study [26] reported a higher outlier rate in raTKA than in mTKA. Three studies reported the incidence of postoperative outliers between raTKA and mTKA [25,26,31]. The lower limb alignment characteristics are reported in Table 3.

**Table 3.** Lower limb alignment characteristics regarding implant position, intraoperative and postoperative measurements, coronal and sagittal alignment, and outliers.

| Author and Publication Year | raTKA N° | Implant Position | Intraoperative and Postoperative Measurements | Coronal and Sagittal Alignment | Outliers > 3° |
|---|---|---|---|---|---|
| Schopper et al. (2022) [21] | 55 | High percentage of outliers in mTKA compared to raTKA after reaching the inflection point | No difference between intraoperative and postoperative measurements | N/A | No outliers >3° (mean angles analyzed within 1°) |
| Thiengwittaya porn et al. (2021) [24] | 75 | N/A | N/A | raTKA improved the postoperative mechanical axis accuracy, sagittal alignment of the femur, and coronal alignment of the tibia | Fewer outliers for HKA, coronal femoral angle, coronal tibial angle, sagittal femoral angle, and sagittal tibial angle in raTKA compared to mTKA |
| Savov et al. (2021) [25] | 59 | The learning curve was not associated with implant position after reaching the inflection point | Significant differences between intraoperative and postoperative measurements (MPTA accurate to 1°, LDFA to 1.6°, and HKA to 2°). | N/A | 5.5% outliers with raTKA on coronal alignment |
| Mahure et al. (2021) [26] | 107 | The learning curve was not associated with implant position after reaching the inflection point | N/A | N/A | Higher outliers for raTKA (43%) than for mTKA (32%) |

**Table 3.** *Cont.*

| Author and Publication Year | raTKA N° | Implant Position | Intraoperative and Postoperative Measurements | Coronal and Sagittal Alignment | Outliers > 3° |
|---|---|---|---|---|---|
| Vermue et al. (2020) [29] | 386 | The learning curve was not associated with implant position after reaching the inflection point | The mean postoperative HKA was 1.2° more valgus than that measured intraoperatively | N/A | N/A |
| Kayani et al. (2018) [33] | 60 | The learning curve was not associated with implant position after reaching the inflection point | N/A | raTKA improved postoperative mechanical axis accuracy, posterior tibial slope, coronal and sagittal alignment of the femur, and coronal and sagittal alignment of the tibia. | N/A |
| Naziri et al. (2019) [32] | 40 | N/A | N/A | No difference in lower limb alignment between raTKA and mTKA | No outliers in both groups |
| Collins et al. (2020) [31] | 72 | N/A | N/A | N/A | 6.9% outliers in coronal plane alignment |

raTKA: robotic-assisted TKA; N°: number of evaluation cases; mTKA: manual TKA; N/A: not available.

*3.4. Complications*

One study reported that one complication was observed during the learning phase (one knee arthrofibrosis). Instead, three complications during the proficiency phase (one knee arthrofibrosis, one deep vein thrombosis, and one surgical site infection) [23]. Three studies reported at least one complication related to raTKA [26,29,33]. Four studies reported no complications related to raTKA [23,27,30,32]. Four studies reported complications not related to raTKA [21,23,28,31]. The most frequently reported complication was reduced postoperative range of motion due to knee arthrofibrosis. The reported complications are summarized in Table 4.

**Table 4.** Learning/Proficiency phase and complications related and not related to raTKA.

| Author and Publication Year | raTKA N° | Learning/Proficiency Phase N° | Complications Related to raTKA N° | Complications Not Related to raTKA N° |
|---|---|---|---|---|
| Schopper et al. (2022) [21] | 55 | N/A | N/A | 1 open patellar tendon rupture, 1 post-traumatic wound dehiscence |
| Mahure et al. (2022) [26] | 107 | N/A | 1 Metallic tack left to the distal femur | N/A |
| Vermue et al. (2022) [29] | 386 | N/A | 1 Diaphyseal femoral stress fracture at pin insertion | N/A |
| Kayani et al. (2019) [33] | 60 | N/A | 1 wound dehiscence at pin insertion | N/A |

**Table 4.** *Cont.*

| Author and Publication Year | raTKA N° | Learning/Proficiency Phase N° | Complications Related to raTKA N° | Complications Not Related to raTKA N° |
|---|---|---|---|---|
| Vanlommel et al. (2021) [23] | 90 | One complication during the learning phase (1 arthrofibrosis). Three complications after reaching the proficiency phase (1 arthrofibrosis, 1 surgical site infection, 1 deep vein thrombosis). | 0 | 2 arthrofibrosis, 1 surgical site infection, 1 deep vein thrombosis |
| Patel et al. (2022) [22] | 604 | N/A | N/A | N/A |
| Bell et al. (2022) [27] | 58 | N/A | 0 | N/A |
| Naziri et al. [32] | 40 | N/A | 0 | N/A |
| King et al. [30] | 202 | N/A | 0 | N/A |
| Ali et al. [28] | 120 | N/A | N/A | 2 arthrofibrosis, 1 cellulitis, 1 acute kidney injury, 1 congestive heart failure |
| Collins et al. [31] | 72 | N/A | N/A | 2 arthrofibrosis, 1 intraoperative tibial periprosthetic fracture, 1 fatal pulmonary embolism |

raTKA: robotic-assisted TKA; N°: number of evaluation cases; N/A: not available.

## 4. Discussion

The use of raTKA is steadily increasing in everyday surgical practice leading to several advantages, such as the reduction of radiographic outliers and risks of iatrogenic soft tissue injuries [14]. Nevertheless, these advantages must be related to higher costs, increased surgical time, and a long learning curve [8,14]. This systematic review aims to evaluate data on the raTKA learning curve in three different areas: surgical time, lower limb alignment, and complications. The main findings of this paper are focused on surgical time. In most cases, the inflection point was recorded between the first 5 and 20 cases, regardless of the surgeon's experience. The first and last cases reported a significant reduction in surgical time [22–28]. Additionally, few studies report the raTKA learning curve regarding lower limb alignment, component position, and intraoperative and postoperative complications [21,23,24].

The main finding of this systematic review is that the inflection points in raTKA range from 5 to 43 cases, with a mean of 14.9 cases. In addition, an average decrease in surgical time of 23.9 min was described between the initial and proficiency phases; the average surgical time in the two phases was 98.8 min and 74.4 min, respectively [21–27]. Schopper et al., in their study, reported that an experienced surgeon could flatten the learning curve, suggesting that companies should provide surgical support from trained personnel during initial cases [21]. Surgical time may be considered an evaluation index of the learning process for surgical procedures. Multiple factors, such as familiarity with displays, landmark registration, and resection techniques, were related to surgical time procedures [38–40]. Several studies included in this review reported improved surgical time in the latter cases compared with the first cases [22–28]. Patel et al. reported that the average surgical time improved significantly between the first and last 50 cases [22].

Similarly, Vanlommel et al. [23] and Bell et al. [27] reported a statistically significant reduction in surgical time in their respective studies. Ali et al. [28] reported a significant improvement in surgical time for only one of the two surgeons involved in their research. Controversially, the results compare the average intervention time of raTKA and mTKA. Thiegwittapown et al. [24] and Naziri et al. [32] reported that the overall time of raTKA was

significantly higher than mTKA. On reaching the inflection point, the average surgical time did not differ between mTKA and raTKA [24,29,35]. Thiegeittaporn et al. [24] observed no difference in the mean surgical time for the last ten raTKA and mTKA cases. Similarly, no surgical time differences were reported both either Ali et al. [28] or Vermue et al. [32] in the last cases of raTKA compared to mTKA.

The robots and the implants used differ, so improvements may be found at different surgery phases [39,41–43]. Kayani et al. [33] reported that the most remarkable improvement was observed in robot setup (from 14.8 min to 9.2 min), bone registration (from 15.8 min to 11.5 min), bone preparation (from 15.8 min to 11.1 min), and joint balancing (from 14.3 min to 9 min). In contrast, no significant improvement was observed in surgical steps, such as surgical approach, implant trial, cement implantation, and closure. Conversely, different data were reported by Vanlommel et al. [23] that noted a significant improvement not only in robotic setup (from 8.9 min to 7.3 min), bone registration (from 103 min to 7.5 min), and joint balancing (from 7 min to 4.1 min) but also for implant trialing (from 17.3 min to 13.5 min). At the same time, no significant improvement was noted for the surgical approach and bone preparation phases. Bell et al. [27] reported a "dramatic increase in efficiency" during the review of the intraoperative phase, in which the surgeon reviews the bone cuts suggested by the robot.

Appropriate component placement in TKA has a crucial role in improving clinical outcomes and patient satisfaction [7,13,14]. Inadequate implant positioning leads to increased polyethylene wear due to higher contact forces and inappropriate soft tissue tension, reducing implant survival [3,5,39]. Some authors have hypothesized that implant malalignment may reduce postoperative satisfaction because of proprioception changes [1,44,45]. Based on this systematic review, most studies did not report an evident learning curve for component positioning accuracy [25–29]. In their paper, Schopper et al. [21] observed a significantly high rate of misaligned components during the raTKA learning phase compared to the proficiency phase. Furthermore, the authors found no statistically significant differences between intraoperative measurement and postoperative values [21]. On the contrary, Savov et al. [25] reported a statistically significant difference between intraoperative and postoperative valuation, and Vermue et al. [29] described a mean postoperative HKA of 1.2° more valgus than for intraoperative. Thiengwittayaporn et al. [24] showed improved accuracy for HKA, femoral sagittal inclination, and tibial coronal inclination in raTKA. Kayani et al. [23] reported better accuracy for HKA, femoral coronal and sagittal alignment, tibial coronal and sagittal alignment, and tibial slope in raTKA. In contrast, Naziri et al. [32] found no statistically significant differences between patients undergoing mTKA and raTKA.

The studies included in this systematic review reported complications both related and unrelated to robotic-assisted surgery. Vermue et al. [29] described persistent tibial pain at the pin placement site. Further inspections revealed a tibial diaphyseal stress fracture caused by the registration pin insertion. The fracture healed without complications in eight weeks. Mahure et al. [26] reported the persistence of a pin metal fragment in the distal femur that did not result in further complications. Vanlommel et al. [23] described a case of arthrofibrosis during both the learning and proficiency curve phases. Additional complications not directly related to robotic-assisted surgery, such as patellar tendon rupture, arthrofibrosis, tibial fracture, surgical wound dehiscence, deep vein thrombosis, and pulmonary embolism, are also described in the studies included in this systematic review [21,23,28,31]. Ra surgery could simplify and minimize the instrumentation needed to perform surgical procedures, significantly reducing instrumentation sterilization and storage costs. In addition, the time required to prepare the operating room would be reduced, optimizing room rotation and equipment costs. Finally, robotic instruments could contribute to the surgeons' training because they provide excellent feedback at the time of use, allow accurate data recording, and offer detailed results analysis at the conclusion of surgery [46].

This systematic review presented some limitations. First, the overall quality of the included studies was low; only two papers had LoE of II, while the other studies had LoEs of III and IV. Second, the surgeons' experiences were very different across the studies, and there was no correlation between these important data and the learning curve. Finally, many papers divided patients into groups, with heterogeneity in the data, including more straightforward cases at the beginning of the learning curve and more complex prostheses in the last phase. Further studies, with a larger and more homogeneous patient sample and longer follow-up, are needed to implement the results obtained in this systematic review.

## 5. Conclusions

Robotic-assisted surgery was introduced into clinical practice recently to improve component positioning and soft tissue balance during TKA. A major concern is related to the length of the learning curve. The most important finding of this systematic review is that the inflection point is reached early, after an average of 14.9 cases. Another significant result is the absence of a learning curve for implant placement and lower limb alignment, meaning that component position is correct from the earliest cases of raTKA. Finally, the current literature has not provided statistically significant data on any difference in the type and rate of complications during the learning curve of raTKA.

**Author Contributions:** G.C., F.B., F.G., S.R. and M.C.: designing the work. G.C. and F.B.: acquisition and analysis of the data. G.C.: drafting the work. G.C., F.B., F.G., A.B. and L.S.: revised the work critically for important intellectual content. S.R., A.M. and L.S.: final approval of the version to be published. All authors have read and agreed to the published version of the manuscript.

**Funding:** This research received no external funding.

**Institutional Review Board Statement:** Not applicable.

**Informed Consent Statement:** Not applicable.

**Data Availability Statement:** No new data were created or analyzed in this study. Data sharing is not applicable to this article.

**Conflicts of Interest:** The authors declare no conflict of interest.

## Abbreviations

| | |
|---|---|
| TKA | total knee arthroplasty |
| raTKA | robotic-assisted TKA |
| n | number |
| PRISMA | Preferred Reporting Items for Systematic Reviews and Meta-Analyses |
| PROSPERO | International Prospective Register of Systematic Reviews |
| MINORS | Methodological Index for Non-Randomized Studies |
| LoE | levels of evidence |
| mTKA | manual TKA |
| MPTA | medial proximal tibial angle |
| LDFA | lateral distal femoral angle |
| HKA | hip–knee–ankle angle |

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
