# Peer review of "Learning Curve in Robotic-Assisted Total Knee Arthroplasty: A Systematic Review of the Literature"

_applsci, doi:10.3390/app122111085_

Round 1
Reviewer 1 Report
This is a Review of manuscript entitled: “Learning curve in robotic-assisted total knee arthroplasty. A systematic review of the literature” by Bosco et al.
Thank you very much for this very interesting paper. It was a pleasure to read such a good systematic review. It is a very modern, quiet unique paper. Learning curve in robotic surgery is very important. We are living in the new era of everything. The progress, as general, in medicine is connected with new technologies. Robotic surgery is a future in all fields of surgery.
My concerns / Suggestions :
Introduction : In my opinion you should extend in introduction section part about robotic surgery. Maybe you should consider adding a new publications in reference section either. I think that you should specify the robotic surgery more widely in other surgical fields as well. Maybe some words about future of robotic surgery in general – please consider summarizing sentences about robotic surgery in discussion section.
Results :
You mentioned as a complication deep vein thrombosis, What was the cause of vein thrombosis in this case ? Was it due to the robotic procedure, or other factors ? Please mention it.
Flow Chart and PRISMA standard: I think that all criteria are done according to the PRISMA standard. Please consider writing more about limitations of the study. In my opinion there is a big limitation of number of studies.
What was the reason of this manuscript ? Are you using surgical robotic assistants ? What are your own experience in this field ? Maybe you should consider your own prospective study. That would be very interesting.
Minor concerns :
Minor English check is required for example :
Line 83 „ igure 1.” Spell check
Table 3 – please extend the description of this table
The same in Table 4 – please extend the description
Thank you very much for this manuscript. I think it should be accepted after revision.
Author Response
Dear Reviewer,
Thank you for your email with the comments. We appreciate the helpful feedback from the reviewers on improving the quality and content of this manuscript. The manuscript has been revised accordingly. We have addressed the specific reviewer concerns in a point-by-point manner below. Changes to the manuscript have been performed and highlighted in yellow. We hope that this revised manuscript is now suitable for publication and look forward to hearing from you.
Best Regards.
Reviewer #1:
This is a Review of manuscript entitled: “Learning curve in robotic-assisted total knee arthroplasty. A systematic review of the literature” by Bosco et al.
Thank you very much for this very interesting paper. It was a pleasure to read such a good systematic review. It is a very modern, quiet unique paper. Learning curve in robotic surgery is very important. We are living in the new era of everything. The progress, as general, in medicine is connected with new technologies. Robotic surgery is a future in all fields of surgery.
My concerns / Suggestions :
1: Introduction : In my opinion you should extend in introduction section part about robotic surgery. Maybe you should consider adding a new publications in reference section either. I think that you should specify the robotic surgery more widely in other surgical fields as well. Maybe some words about future of robotic surgery in general – please consider summarizing sentences about robotic surgery in discussion section.
- 1: As requested, the suggested change has been made.
Please see edited text (Introduction section):
“Ra surgery has experienced a great fascination since its beginnings around 1980. Ra procedures allow surgeons to perform complex operations with greater precision, flexibility, and control than conventional techniques and have become a staple in operating rooms worldwide. In recent decades, considerable technological advances have made robotics applications possible in multiple surgery fields, including orthopedics [12]. Robotic-assisted TKA (raTKA) has gained popularity, and its use is steadily growing [13,14].”
Please see edited text (Discussion section):
“Ra surgery could simplify and minimize the instrumentation needed to perform surgical procedures, significantly reducing instrumentation sterilization and storage costs. In addition, the time required to prepare the operating room would be reduced, optimizing room rotation and equipment costs. Finally, robotic instruments could contribute to the surgeons' training because they provide excellent feedback at the time of use, allow accurate data recording, and detailed results analysis at the surgery conclusion [46].”
--
2: Results:
You mentioned as a complication deep vein thrombosis, What was the cause of vein thrombosis in this case? Was it due to the robotic procedure, or other factors? Please mention it.
- 2: "The reported case of deep vein thrombosis was not related to raTKA. The authors (Vanlommel et al.) reported this complication in the postoperative period without specifying a specific cause. We reported this finding in Table 4 under <<Complications not related to raTKA>>."
--
3: Flow Chart and PRISMA standard: I think that all criteria are done according to the PRISMA standard. Please consider writing more about limitations of the study. In my opinion there is a big limitation of number of studies.
- 3: As requested, the suggested change has been made.
Please see edited text:
“Further studies, with a larger and more homogeneous patient sample and longer follow-up, are needed to implement the results obtained in this systematic review.”
--
4: What was the reason of this manuscript? Are you using surgical robotic assistants? What are your own experience in this field? Maybe you should consider your own prospective study. That would be very interesting.
- 4: Thank you for the comment. As reported in the introduction and discussion session, this manuscript aims to analyze the surgical time learning curve in raTKA. We use two types of robots for TKA (Navio and ROSA). To date, we have not yet started a prospective or retrospective study focusing on our case series of raTKA. Therefore we have not included our case series in this systematic literature review. As a next step, we greatly appreciate your suggestion. We will consider conducting a study focusing on our case history of raTKA as soon as we have an adequate number of cases.
--
5: Minor concerns:
Minor English check is required for example :
Line 83 „ igure 1.” Spell check
- 5: As requested, the suggested change has been made.
--
6: Table 3 – please extend the description of this table
The same in Table 4 – please extend the description
- 6: As requested, the suggested change has been made.
Please see edited text:
“Table 3. Lower limb alignment characteristics regarding implant position, intraoperative and postoperative measurements, coronal and sagittal alignment, and outliers.”
“Table 4. Learning/Proficiency phase and complications related and not related to raTKA.”
--
7: Thank you very much for this manuscript. I think it should be accepted after revision.
- 7: Thanks for the comment.

Reviewer 2 Report
I thank the authors for allowing me to evaluate this work which is surely very relevant for orthopaedists. From a methodological point of view, this work is rigorous and well conducted. The description of the results is easily readable and well supported by tables. The discussion is in line with the current state of the art in this type of surgery.
I suggest only two small corrections: Line 24 "TKR." and line 74, there is a problem with punctuation and quotation marks. I think the authors wanted to say "TKR".
As it is, the article is quite publishable in Applied Sciences.
Author Response
Dear Reviewer,
Thank you for your email with the comments. We appreciate the helpful feedback from the reviewers on improving the quality and content of this manuscript. The manuscript has been revised accordingly. We have addressed the specific reviewer concerns in a point-by-point manner below. Changes to the manuscript have been performed and highlighted in yellow. We hope that this revised manuscript is now suitable for publication and look forward to hearing from you.
Best Regards.
Reviewer #2:
I thank the authors for allowing me to evaluate this work which is surely very relevant for orthopaedists. From a methodological point of view, this work is rigorous and well conducted. The description of the results is easily readable and well supported by tables. The discussion is in line with the current state of the art in this type of surgery.
1: I suggest only two small corrections: Line 24 "TKR." and line 74, there is a problem with punctuation and quotation marks. I think the authors wanted to say "TKR".
- 1: As requested, the suggested change has been made. Please see edited text.
2: As it is, the article is quite publishable in Applied Sciences. - 2: Thanks for the comment.

Reviewer 3 Report
The paper entitled, the Learning curve in robotic-assisted total knee arthroplasty: A systematic review of the literature. Reviewer comments are as the following:
Abstract
1. Author needs to address the main finding of the review clearly.
Material and Methods
2. Line 64, It is suggested to write a brief paragraph to introduce sections 2.1, 2.2, 2.3 ...
3. Figure 1 line 101, Justify whether n = 15 is significant statistically.
Discussion
4. The discussion section is acceptable and quite comprehensive.
Conclusions
5. The conclusion is clear.
6. It is suggested to add the future direction of the study.
References
7. The references are up to date.

Author Response
Dear Reviewer,
Thank you for your email with the comments. We appreciate the helpful feedback from the reviewers on improving the quality and content of this manuscript. The manuscript has been revised accordingly. We have addressed the specific reviewer concerns in a point-by-point manner below. Changes to the manuscript have been performed and highlighted in yellow. We hope that this revised manuscript is now suitable for publication and look forward to hearing from you.
Best Regards.
Reviewer #3:
The paper entitled, the Learning curve in robotic-assisted total knee arthroplasty: A systematic review of the literature. Reviewer comments are as the following:
Abstract
- Author needs to address the main finding of the review clearly.
- 1: As requested, the suggested change has been made.
Please see edited text:
“The main finding of this systematic review is that the procedure number required to reach the learning plateau is about 14.9 cases. Furthermore, an average decrease in surgical time of 23.9 minutes was described between the initial and proficiency phases; the average surgical time in the two phases was 98.8 minutes and 74.4 minutes, respectively.”
--
Material and Methods
- Line 64, It is suggested to write a brief paragraph to introduce sections 2.1, 2.2, 2.3 ...
- 2: As requested, the suggested change has been made.
I have written the paragraph provided below to introduce the subsequent Sections 2.1, 2.2, 2.3 ...
Please see edited text:
“A systematic literature review was performed according to the Preferred Reporting Items for Systematic Reviews and Meta-Analyses (PRISMA) guidelines [18]. The study was registered on the International Prospective Register of Systematic Reviews (PROSPERO), ID: CRD42022354797, in August 2022 [19,20].”
--
- Figure 1 line 101, Justify whether n = 15 is significant statistically.
- 3: There is no minimum number to be included in a systematic review. However, the greater the number of studies, the greater the chances of conducting a meaningful analysis. I attach the DOI of a recent systematic review with three studies included as an example of systematic review (doi: 10.1016/j.ortho.2021.01.004.). Including 15 studies in our systematic review greatly reduced the risk of methodological bias (performed with the MINORS score), but assessing statistical significance regarding the number of included studies in a systematic review is not applicable.
--
Discussion
- The discussion section is acceptable and quite comprehensive.
- 4: Thanks for the comment.
--
Conclusions
- The conclusion is clear.
- 5: Thanks for the comment.
--
- It is suggested to add the future direction of the study.
- 6: As requested, the suggested change has been made.
Please see edited text (Discussion section):
“Ra surgery could simplify and minimize the instrumentation needed to perform surgical procedures, significantly reducing instrumentation sterilization and storage costs. In addition, the time required to prepare the operating room would be reduced, optimizing room rotation and equipment costs. Finally, robotic instruments could contribute to the surgeons' training because they provide excellent feedback at the time of use, allow accurate data recording, and detailed results analysis at the surgery conclusion [46].”
--
References
- The references are up to date.
- 7: Thanks for the comment.
